# Smaller Models are Natural Explorers for Policy-Level Diversity in GRPO

Yiming Ren [* 1 2]  Yiran Xu [* 1]  Zicheng Lin [* 1]  Chufan Shi [1]  Yukang Chen [3]
Dingdong Wang [3]  Tianhe Wu [4]  Junjie Wang [1]  Yujiu Yang [1]  Yu Qiao [2]  Ruihang Chu [1]

## Abstract

We identify a new dimension for enhancing roll-out diversity in Group Relative Policy Optimization (GRPO) for LLMs. While GRPO relies on diverse rollouts, prevailing strategies primarily increase diversity by injecting more token-level randomness, which may introduce step-wise noise and lead to incoherent trajectories. We uncover that smaller models within the same model family inherently exhibit higher policy-level diversity, indicated by their superior pass@k relative to larger counterparts as sample counts increase. Unlike token-level noise, this diversity is temporally correlated, preserves logical consistency, and provides structured exploration signals for gradient estimation. We thus propose S2L-PO (Small-to-Large Policy Optimization), a framework that leverages fixed small models as natural explorers to train larger models. To balance exploration and exploitation, we design a progressive annealing strategy that transitions from offline small-model rollouts to the large learner's own sampling. This shift elegantly avoids mid-training performance drops caused by the small model's capacity limits, achieving faster convergence and unlocking a higher performance ceiling. S2L-PO improves accuracy on diverse mathematical reasoning benchmarks (*e.g.*, +8.8% on AIME 24 using a 1.7B explorer to guide the 8B model) while reducing rollout compute.

## 1. Introduction

Reinforcement learning with verifiable rewards (RLVR) has emerged as a powerful paradigm for improving the reasoning capabilities of large language models (Guo et al., 2025;

Hong et al., 2024; Wang et al., 2025d). Group Relative Policy Optimization (GRPO) (Shao et al., 2024), in particular, has gained widespread adoption due to its simplicity and effectiveness: it samples multiple candidate solutions per prompt, computes group-relative advantages, and updates the policy without requiring a separate critic network. A key factor in GRPO's success is the diversity of sampled rollouts. When candidates within a group are too homogeneous, the advantage signals collapse and learning stagnates (Gu et al., 2025; Wang et al., 2025d; Zhang et al., 2025b).

Prevailing strategies for increasing rollout diversity primarily operate on the token level. A common approach is temperature scaling, which raises the original sampling temperature to inject more randomness into individual token selection. Yet, high-temperature sampling can trigger entropy explosion (Nguyen et al., 2024; Shi et al., 2024b; Wang et al., 2025c; Yang et al., 2025b; Zhuang et al., 2025), where the policy explores indiscriminately across all tokens, leading to training instability and degraded reasoning performance. More critically, because elevated temperature adds randomness independently at each decoding step, small deviations compound over long reasoning chains, making it difficult to maintain a consistent logical flow. Such resulting rollouts may exhibit high surface diversity in terms of token entropy but often suffer from low behavioral coherence. Ultimately, this approach is less effective at providing the structured exploration signals that GRPO requires. While other works explore curating diverse response sets to improve training signals or rewarding intra-group diversity (Anschel et al., 2025; Chen et al., 2025), these strategies involve data engineering and extra computational overhead, limiting their scalability to new tasks without significant costs.

We present an empirical finding to explore an alternative dimension for enhancing diversity. When comparing models of different sizes on mathematical reasoning benchmarks, we observe a surprising pattern: while larger models outperform their smaller counterparts at pass@1, this gap shrinks and can even reverse as $k$ increases (see Fig. 2). For instance, a 4B model surpasses an 8B model in pass@$k$ once $k \geq 32$, and it can also outperform a 14B model when the sample budget is sufficiently large (*e.g.*, $k \approx 200$). As smaller models have a lower performance floor, their competitiveness or advantage at higher sample counts suggests that they pos-

[1]Tsinghua University [2]Shanghai AI Laboratory [3]The Chinese University of Hong Kong [4]City University of Hong Kong. Correspondence to: Yu Qiao <qiaoyu@pjlab.org.cn>, Ruihang Chu <ruihangchu@mail.tsinghua.edu.cn>.

*Proceedings of the 43rd International Conference on Machine Learning*, Seoul, South Korea. PMLR 306, 2026. Copyright 2026 by the author(s).

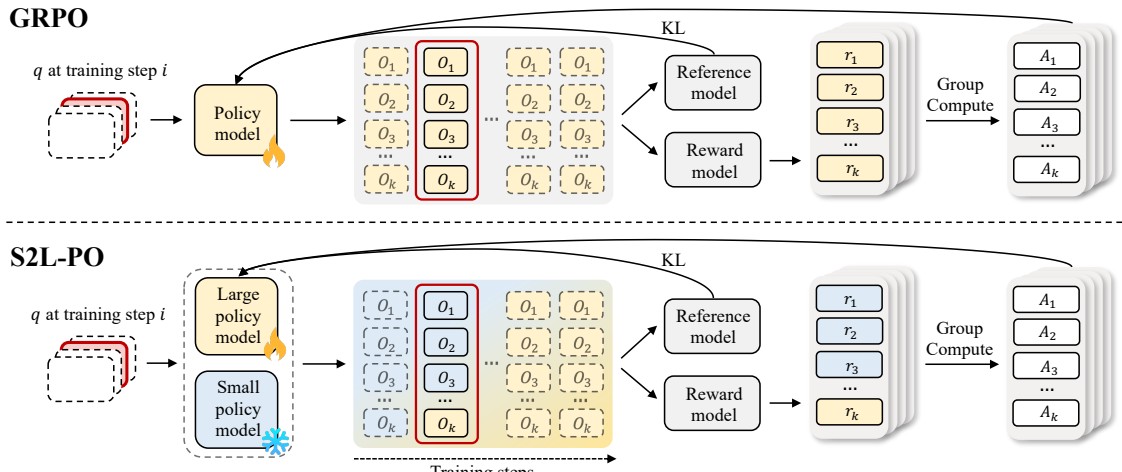

*Figure 1.* S2L-PO (**Bottom**) simply modifies the rollout generation process of standard GRPO (**Top**). Motivated by the observation that smaller models inherently exhibit higher policy-level diversity, S2L-PO leverages a frozen smaller policy model to sample diverse rollouts for training a larger model. In early training, rollouts are primarily sampled from the smaller model to encourage diverse exploration. As training progresses, sampling smoothly transitions through a mixture of smaller and larger models, and ultimately recovers standard on-policy GRPO to balance exploration and exploitation.

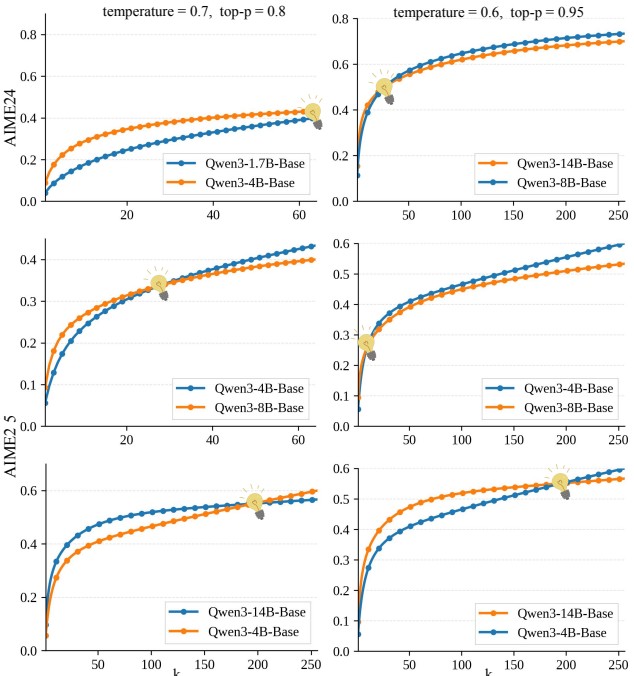

*Figure 2.* Pass@k curves on AIME24 and AIME25 for Qwen3 Base models of various scales. While larger models perform better at small $k$, smaller models continue to improve as $k$ increases and can match or exceed larger models under large sampling size.

sess an inherent diversity, stemming not from token-level randomness but from more varied solution strategies (Bansal et al., 2024; Dragoi et al., 2025; Yue et al., 2025).

We characterize this phenomenon as a form of policy-level diversity. Smaller models typically undergo distillation from larger models within the same family, ensuring distributional alignment while reducing parameter count. This parameter-

level compression induces a structured shift in the policy's inductive bias. Unlike token-level diversity that perturbs the action distribution through step-wise noise along a reasoning trajectory, parameter-level compression applies a time-invariant perturbation to shift the entire policy. As analyzed in Sec. 3.1, this preserves temporal correlation and enhances internal consistency. It further prevents gradient dilution by focusing exploration on structured reasoning strategies rather than uncoordinated local flips, providing more informative updates for the learner. We summarize this distinction as follows: *token-level randomness perturbs the action; parameter-level compression perturbs the policy.*

Building on this insight, we propose S2L-PO (Small-to-Large Policy Optimization), a framework that leverages weaker small models as natural explorers to generate rollouts for training stronger large models (see Fig. 1). Since the small model can provide superior policy diversity per compute unit, we fix its parameters to generate rollouts offline. This setting avoids the instability of early-stage on-policy updates caused by mismatched model capacities and enables highly efficient parallelization of rollout generation. To balance exploration and exploitation, we design a progressive annealing strategy that transitions from small-model exploration to on-policy learning. Initially, the small model provides the entire or the majority of rollouts to maintain diverse exploration and prevent mode collapse. As training progresses, we gradually shift the sampling role to the large model to mitigate the distribution mismatch between the small sampler and the large learner. This approach effectively prevents mid-training performance degradation and ultimately achieves a higher ceiling. Since S2L-PO only modifies the rollout process, it remains seamlessly compatible with existing GRPO implementations.

We comprehensively evaluate our approach across two model families (Qwen3 and InternLM2.5) on four mathematical reasoning benchmarks (AIME24, AIME25, MATH-500, and OlympiadBench). Across various settings, small-to-large policy sampling consistently improves both final performance and sample efficiency over standard GRPO, reaching stronger Pass@1 with fewer effective training steps (*e.g.*, using a 1.7B explorer to guide an 8B model yields an average gain of about 9%). On an out-of-domain benchmark (CommonsenseQA), our method matches or marginally improves over GRPO, suggesting that the benefits do not come at the expense of generalization. Code is available at `https://github.com/qishisuren123/S2L-PO`.

## 2. Preliminary

### 2.1. Group Relative Policy Optimization (GRPO)

Group Relative Policy Optimization (GRPO) (Shao et al., 2024) is an on-policy policy-gradient method tailored to RLVR settings, where supervision is provided by *verifiable* rewards (*e.g.*, rule-based correctness checks). GRPO optimizes a policy model $\pi_\theta$ without training an explicit value function (critic). Instead, it estimates advantages via within-group relative comparisons among multiple samples generated for the same query, which reduces both compute and engineering overhead.

Formally, for each query $q \sim \mathcal{D}$, GRPO samples a group of $k$ candidate outputs $O = \{o_1, o_2, \ldots, o_k\}$ from the behavior policy $\pi_{\theta_{\text{rollout}}}$ and evaluates each output with a scalar reward $r(o_i)$. It then computes a *group-relative advantage* by standardizing rewards within the sampled group:

$$A_i = \frac{r(o_i) - \text{mean}(\{r(o_j)\}_{j=1}^k)}{\text{std}(\{r(o_j)\}_{j=1}^k) + \epsilon_{\text{adv}}}. \tag{1}$$

Let $\rho_i = \frac{\pi_\theta(o_i|q)}{\pi_{\theta_{\text{rollout}}}(o_i|q)}$ denote the importance sampling ratio. GRPO uses a PPO-style clipped surrogate objective with KL regularization toward a reference policy $\pi_{\text{ref}}$. We optimize $\mathcal{J}_{\text{GRPO}}(\theta)$ defined as:

$$\mathbb{E}\left[\frac{1}{k}\sum_{i=1}^k \min\left(\rho_i A_i,\ \text{clip}(\rho_i, 1 - \epsilon_{\text{clip}}, 1 + \epsilon_{\text{clip}})A_i\right)\right.$$
$$\left. - \beta\, \mathbb{D}_{\text{KL}}(\pi_\theta \,\|\, \pi_{\text{ref}})\right]. \tag{2}$$

Since GRPO relies on within-group relative rewards, its gradient quality is sensitive to the diversity of sampled candidates. When candidates are homogeneous, advantage signals vanish and learning stagnates. The standard remedy, temperature scaling, injects token-level noise that is temporally independent, often yielding locally random but globally incoherent trajectories. This motivates our exploration of policy-level perturbations as an alternative source of structured diversity.

### 2.2. Distillation Introduces Perturbations

In this work, we study the parameter-count compression within a single model family and its implications for exploration in RL-style fine-tuning. Rather than viewing compression purely as an efficiency tool, we interpret the compression-and-distillation process as inducing a policy-level perturbation (Gu et al., 2024; Hinton et al., 2015; Park & Cho, 2025; Peng & Zhang, 2025). Although compression is often motivated by deployment constraints (*e.g.*, memory, latency, and serving cost), the student is typically optimized to retain the teacher's task behavior under a reduced parameter budget. As a result, the teacher-to-student mapping is not arbitrary: it induces a structured shift in inductive biases and decision boundaries. We leverage this property and treat the compressed student as a coherent deviation from its teacher in policy space, providing a source of exploration diversity.

Take Qwen3 dense series (Yang et al., 2025a) as example, which represents a currently strong and widely adopted base model family. For models at or below 14B parameters, they are obtained via a unified larger-to-smaller distillation in final stages. As reported, each model is trained as the student to align its logits to those of a larger teacher (*e.g.*, Qwen3-32B or Qwen3-235B-A22B) by minimizing a KL-divergence objective during on-policy distillation. This yields a controlled compression setting: students across scales share a consistent distillation procedure and teacher family, ensuring behavioral proximity while allowing capacity reduction to induce meaningful, structured deviations.

Formally, let $\pi_{\theta^\star}$ denote the teacher policy and $\{\pi_{\theta_k}\}_{k \in \mathcal{K}}$ denote student policies at different parameter scales within the same series, where $\mathcal{K} = \{1.7B, 4B, 8B, 14B\}$. Since $\pi_{\theta_k}$ is trained to approximate $\pi_{\theta^\star}$ under distillation, we model compression as an effective perturbation in parameter space, formulated as

$$\theta_k \approx \theta^\star + \delta_{\theta,k}, \tag{3}$$

where $\delta_{\theta,k}$ captures the structured change induced by compression and distillation. Equivalently, this corresponds to a controlled deviation in policy space:

$$\pi_{\theta_k}(\cdot \mid q) \approx \pi_{\theta^\star}(\cdot \mid q) + \Delta_{\pi,k}(\cdot \mid q), \tag{4}$$

where $\Delta_{\pi,k}$ is a *coherent* shift arising from reduced capacity, rather than a token-level random perturbation. In this paper, we validate the perturbation view on both Qwen3 (Yang et al., 2025a) and InternLM2.5 (Cai et al., 2024) families.

# 3. Method

Given that compression induces structured, temporally consistent perturbations, we first analyze why policy-level perturbations yield new kinds of exploration signals compared to token-level noise (Section 3.1). Then we present S2L-PO framework that leverages this property (Section 3.2).

## 3.1. Token-Level vs. Policy-Level Perturbations

Given a policy and a query, GRPO samples a group of rollouts and constructs group-relative advantages for policy updates. The exploration mechanism determines how these rollouts deviate from each other in policy space and directly influences the quality of gradient estimates. In standard GRPO implementations, rollouts are sampled with a modest non-zero temperature to balance training stability and within-group diversity, which already introduces a baseline level of token-level randomness. In this paper, we treat this default temperature as part of the GRPO baseline and focus on *additional* sources of diversity beyond it.

**Token-level perturbations.**   It refers to introducing additional step-wise randomness in action selection beyond the baseline sampling temperature used in GRPO. A typical instance is sampling from a softened distribution

$$a_t \sim \pi^{\text{tok}}(\cdot \mid s_t),\ \pi^{\text{tok}}(a \mid s_t) = \frac{\exp(l_t(a)/T)}{\sum_{a'} \exp(l_t(a')/T)}, \tag{5}$$

where $l_t(\cdot)$ denotes the logits at step $t$ and $T$ controls the perturbation strength. Equivalently, this process can be expressed using the Gumbel–Max formulation

$$a_t = \arg\max_a \big(l_t(a)/T + \epsilon_t(a)\big),\ \{\epsilon_t(\cdot)\}_{t\geq 1}\ \text{i.i.d.,} \tag{6}$$

where the noise sources are independent across steps. Importantly, while the injected noise sources are i.i.d., the realized tokens $\{a_t\}$ are generally not i.i.d. because the state $s_t$ depends on previous actions.

Token-level diversification draws actions from a perturbed conditional distribution at each step, $a_t \sim \pi^{\text{tok}}(\cdot \mid s_t)$. Let $o = (a_1, \ldots, a_L)$ be the resulting sequence and define the *prefix match event*

$$M_t := \mathbb{I}\{(a_1, \ldots, a_t) = (a_1^\star, \ldots, a_t^\star)\}, \tag{7}$$

where $o^\star$ denotes a deterministic reference decoding trace under the base policy, used only to define whether a rollout remains on the same decision path. For any step $t$,

$$\Pr(M_t = 1) = \prod_{j=1}^{t} \Pr\big(a_j = a_j^\star \mid M_{j-1} = 1\big). \tag{8}$$

Moreover, consider a regime where token-level randomness is increased relative to the GRPO baseline, so that the per-step deviation probability admits a lower bound $p > 0$ over

the considered horizon, *i.e.*,

$$p \leq \Pr\big(a_j \neq a_j^\star \mid M_{j-1} = 1\big) \quad j \in \{1, \ldots, t\}, \tag{9}$$

$$\Pr(M_t = 1) \leq (1 - p)^t, \tag{10}$$

so the mass of trajectories that share a common early prefix decays exponentially with $t$.

This decay implies that for long-horizon outputs, late tokens are increasingly generated under a mixture over divergent prefixes. A convenient proxy for the resulting loss of temporal dependence is the growth of $\Pr(M_t = 0)$. In particular, for bounded features $f(a_t)$ and $g(a_s)$ with $\|f\|_\infty, \|g\|_\infty \leq 1$ and $t < s$, one can obtain under a mild mixture/coupling assumption a problem-dependent constant $C > 0$ such that

$$\big|\text{Cov}(f(a_t), g(a_s) \mid q)\big| \leq C \Pr(M_t = 0). \tag{11}$$

Thus, as $\Pr(M_t = 0)$ grows with $t$ for long generations, long-range cross-token dependence weakens, making earlier and later decisions less mutually consistent.

**Policy-level perturbations via parameter-level compression.**   In contrast, parameter-level compression (in this work, primarily via distillation to a smaller model) induces an effective structured perturbation in parameter space. We abstract this effect by an equivalent additive perturbation

$$\tilde{\theta} = \theta + \delta_\theta,\ a_t \sim \pi_{\tilde{\theta}}(\cdot \mid s_t), \tag{12}$$

where $\delta_\theta$ represents a time-invariant modification of the policy parameters during the rollout. Although the resulting logit shifts depend on context through the forward pass, all steps share the same perturbed policy $\pi_{\theta + \delta_\theta}$.

For any fixed state $s$, define the local distributional shift

$$\Delta \pi_s(a) := \pi_{\theta + \delta_\theta}(a \mid s) - \pi_\theta(a \mid s). \tag{13}$$

Since the same $\delta_\theta$ is applied at every step, the induced shifts $\{\Delta \pi_{s_t}\}_{t=1}^{L}$ are coupled through shared parameters, yielding trajectory-level deviations that are typically temporally correlated rather than independent across $t$. Intuitively, this correlation encourages trajectories to follow a coherent alternative strategy throughout the rollout generation.

**Implications for gradient estimation in GRPO.**   We now examine how these differences affect policy-gradient estimates. For a sampled trajectory $o_i$ with group-relative advantage

$$A_i = \frac{r_i - \mu_r}{\sigma_r}, \tag{14}$$

and the GRPO policy-gradient contribution is

$$\mathbf{g}_i = A_i \nabla_\theta \log \pi_\theta(o_i \mid q) = A_i \sum_{t=1}^{L} \nabla_\theta \log \pi_\theta(a_{i,t} \mid s_{i,t}). \tag{15}$$

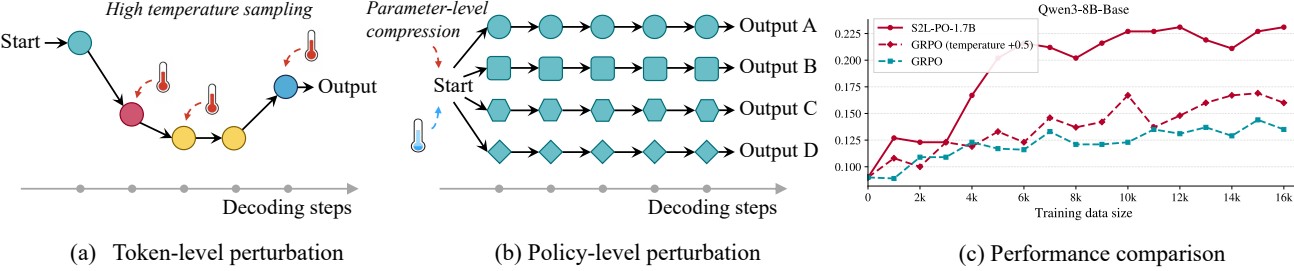

*Figure 3.* Two ways to increase rollout diversity under standard GRPO. (a) Increasing token-level perturbation (*e.g.*, higher sampling temperature) introduces step-wise stochasticity that accumulates over decoding steps, often reducing long-range coherence. (b) Policy-level perturbations (*e.g.*, parameter-level compression within a model family) induce temporally consistent trajectory deviations, yielding diverse yet structured policy paths. (c) Increasing token-level randomness beyond the GRPO baseline yields limited gains, whereas adding policy-level diversity enables more effective exploration and significantly better results. Blue thermometers indicate the default GRPO temperature; red indicate higher temperatures. Different colors denote token-level diversity; different shapes denote policy-level diversity.

Let $\mathbf{u}_{i,t} := \nabla_\theta \log \pi_\theta(a_{i,t} \mid s_{i,t})$ denote the per-step score function, so $\mathbf{g}_i = A_i \sum_{t=1}^{L} \mathbf{u}_{i,t}$. Since $A_i$ is a scalar shared across steps, it scales squared norms by $A_i^2$ but does not change the cross-step interference mechanism; we therefore analyze $\sum_{t=1}^{L} \mathbf{u}_{i,t}$ for clarity.

**Gradient interference under token-level perturbations.**
Expanding the squared norm gives

$$\left\| \sum_{t=1}^{L} \mathbf{u}_{i,t} \right\|^2 = \sum_{t=1}^{L} \|\mathbf{u}_{i,t}\|^2 + 2 \sum_{1 \leq t < s \leq L} \langle \mathbf{u}_{i,t}, \mathbf{u}_{i,s} \rangle. \quad (16)$$

Under strengthened token-level perturbations, prefix divergence suppresses long-range dependence (cf. (11)). Concretely, given bounded scalar projections $z_{i,t} := \langle \mathbf{u}_{i,t}, \mathbf{v} \rangle$ with $\|\mathbf{v}\| = 1$, there exist tasks such that

$$\left| \mathrm{Cov}(z_{i,t}, z_{i,s} \mid q) \right| \leq c \Pr(M_t = 0), \qquad t < s, \quad (17)$$

where $c > 0$ is a constant depending on the bound of $z_{i,t}$. A formal proof via the law of total covariance, which eliminates the need of coupling assumptions and yields $c = 5B^2$ with $B$ bounding $|z_{i,t}|$, is given in Proposition E.2 (Appendix E). As $\Pr(M_t = 0)$ grows with $t$ in long outputs, correlations between distant steps are suppressed. Consequently, the large-lag cross terms in Eq. (16) are less coherent and tend to cancel in expectation, so the accumulation behaves closer to a random-walk sum when long-range alignments vanish:

$$\mathbb{E}\left[ \left\| \sum_{t=1}^{L} \mathbf{u}_{i,t} \right\|^2 \right] \approx \sum_{t=1}^{L} \mathbb{E}\left[ \|\mathbf{u}_{i,t}\|^2 \right] \quad (18)$$

For long-horizon reasoning (*e.g.*, mathematical solution generation), this implies weaker cross-step reinforcement: per-token score contributions are less consistently aligned across the horizon, which can make the trajectory-level update direction noisier and less stable under GRPO.

**Structured gradients under policy-level perturbations.**
For a fixed parameter-level perturbation $\delta_\theta$ shared across the rollout, a local expansion yields, for any fixed trajectory,

$$\nabla_\theta \log \pi_{\theta+\delta_\theta}(o \mid q) \approx \nabla_\theta \log \pi_\theta(o \mid q) + \nabla_\theta^2 \log \pi_\theta(o \mid q) \, \delta_\theta. \quad (19)$$

Although Eq. (19) is local, it highlights a key qualitative difference: the rollout is generated under a single, consistently shifted policy, which tends to induce temporally consistent deviations throughout the trajectory. As a result, per-step score contributions are more likely to remain aligned across time, increasing cross-step reinforcement in Eq. (16) and mitigating the long-lag cancellation behavior implicit in Eq. (18). In long-horizon settings, this yields a more coherent trajectory-level gradient signal for GRPO group comparisons.

More formally, we show in Proposition E.4 (Appendix E) that the cross-step covariance under policy-level perturbation admits a *positive lower bound*:

$$|\mathrm{Cov}(\tilde{z}_{i,t}, \tilde{z}_{i,s} \mid q)| \geq \gamma - 5B^2 \Pr_{\mathrm{std}}(M_t = 0) - O(\|\delta_\theta\|^3), \quad (20)$$

where $\gamma := \mathbb{E}[\mathbf{v}^\top H_t \Sigma_\delta H_s^\top \mathbf{v} \mid q] > 0$ captures Hessian alignment under the parameter perturbation $\delta_\theta$, and $\Pr_{\mathrm{std}}$ is evaluated at the standard (unperturbed) temperature. Unlike the token-level upper bound in Eq. (17) that becomes vacuous as $\Pr(M_t = 0) \to 1$, this lower bound remains positive when the Hessian alignment $\gamma$ is sufficiently large, ensuring constructive gradient interference across steps.

**Takeaway.** Token-level randomness can accumulate over decoding steps, which may break long-range coherence and increase gradient interference as shown in Eqs. (16)–(18); policy-level perturbations induce time-correlated deviations that preserve coherence and yield more structured GRPO gradients. Fig. 3 illustrates these two different ways by showcasing their mechanisms, as well as their actual impact on model performance.

## 3.2. S2L-PO: Small-to-Large Policy Optimization

Guided by the above analysis, we propose S2L-PO, a framework that leverages smaller (compressed) models for exploration while training larger models for exploitation. The core idea is simple: since smaller models provide richer behavioral diversity per unit compute, we use them to generate rollouts and train the larger policy via GRPO. The complete procedure is summarized in Algorithm 1.

**Mixed rollout generation.** At each training step, we construct a mixed rollout distribution. Given a group size $G$, we sample $G_w$ candidates from a frozen smaller policy $\pi_\omega$ and $G_s = G - G_w$ candidates from the trainable larger policy $\pi_\theta$. The smaller policy remains frozen throughout training and serves solely as an exploration agent. The larger policy is updated using GRPO with group-relative advantages computed over the combined candidate set.

**Progressive annealing.** We linearly anneal the smaller-to-larger ratio over the first $T_{\mathrm{mix}}$ training steps. In our implementation, $T_{\mathrm{mix}}$ defaults to the first half of the total training process. Early in training, when the larger policy is unstable and prone to mode collapse, the smaller model provides diverse exploration at low cost, alleviating vanishing advantage signals. As training converges, reducing $G_w$ mitigates distribution mismatch, ensuring the final policy is optimized under its own behavior. After step $T_{\mathrm{mix}}$, the framework recovers standard on-policy GRPO.

**Compatibility and efficiency.** S2L-PO does not modify the GRPO objective, advantage construction, or optimization procedure; it only changes how rollouts are generated. As a result, it can be plugged into existing GRPO pipelines with minimal engineering effort and remains orthogonal to complementary techniques such as reward shaping or curriculum learning. In addition, using a smaller rollout policy reduces the per-sample generation cost, and the same weak-model rollouts can be reused across multiple strong-model training runs, further amortizing rollout compute. Since rollout generation is typically the dominant time and compute bottleneck in GRPO, these properties translate into direct savings in FLOPs and wall-clock time, and in principle can shorten end-to-end training by reducing the rollout burden.

## 4. Experiment

### 4.1. Experiment Settings

We train on the deduplicated DAPO17k (Yu et al., 2025) focusing on verifiable multi-step reasoning. For evaluation we choose four mathematical reasoning benchmarks: AIME 2024, AIME 2025 (Balunović et al., 2025), MATH-500 (Hendrycks et al., 2021), and OlympiadBench (He et al., 2024), and additionally report out-of-domain (OOD) gen-

---

**Algorithm 1** S2L-PO: GRPO with Progressive smaller-to-larger Rollout Sampling

---

**Require:** Trainable policy $\pi_\theta$, frozen smaller policy $\pi_\omega$, reward function $r_\phi$, prompt dataset $\mathcal{D}$, group size $G$, total training steps $T$, transition step $T_{\mathrm{mix}}$, GRPO update steps per iteration $U$
**Ensure:** Optimized policy $\pi_\theta$
1: **for** $i = 1$ to $T$ **do**
2:     Sample a batch of prompts $\mathcal{D}_b \subset \mathcal{D}$.
3:     **if** $i \leq T_{\mathrm{mix}}$ **then**
4:         {Progressive smaller-to-larger rollout phase}
5:         $\alpha \leftarrow 1 - \frac{i-1}{T_{\mathrm{mix}}-1}$
6:         $G_w \leftarrow \lceil \alpha G \rceil, G_s \leftarrow G - G_w$
7:     **else**
8:         {Pure on-policy GRPO phase}
9:         $G_w \leftarrow 0, G_s \leftarrow G$
10:     **end if**
11:     **for all** $q \in \mathcal{D}_b$ **do**
12:         **if** $G_w > 0$ **then**
13:             Sample $G_w$ candidates from $\pi_\omega(\cdot \mid q)$
14:         **end if**
15:         Sample $G_s$ candidates from $\pi_\theta(\cdot \mid q)$
16:         Form candidate group $\mathcal{O}(q)$
17:     **end for**
18:     Compute rewards using $r_\phi$ and group-relative advantages following GRPO
19:     **for** $u = 1$ to $U$ **do**
20:         Update $\theta$ by maximizing the GRPO objective
21:     **end for**
22: **end for**
23: **return** $\pi_\theta$

---

eralization on CommonsenseQA (Talmor et al., 2019). All evaluations are in `nothink` mode following the Qwen3 technical report (Yang et al., 2025a). We sample 16 rollouts per question and compute Pass@1 by averaging the per-problem success indicator over the dataset. To demonstrate cross-family generalizability, we evaluate on two model families: Qwen3-Base (Yang et al., 2025a) and InternLM2.5-Base (Cai et al., 2024). For Qwen3, the 1.7B and 4B variants serve as smaller rollout actors for 8B and 14B target policies. For InternLM2.5, the 1.8B model serves as explorer for the 7B target. All runs are conducted on a single node with 8 NVIDIA L20 GPUs using the default GRPO configuration in `verl` (Sheng et al., 2024).

### 4.2. Main Results

**Small-to-large sampling improves both convergence speed and final performance.** As illustrated in Fig. 3a and Fig. 3b, our approach leverages a smaller model to introduce policy-level diversity. Fig. 3c contrasts this with increasing token-level noise (Temperature = 1.5). Unlike high-temperature sampling, which suffers from instability

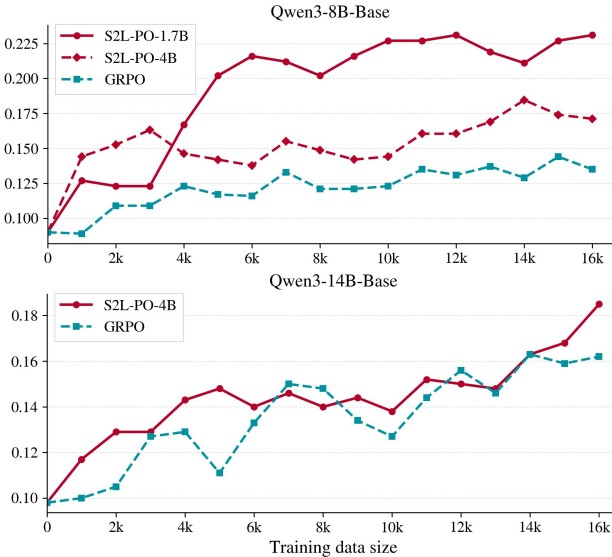

*Figure 4.* S2L-PO improves both final performance and convergence speed. Pass@1 on AIME24&25 versus effective training progress for different scale transitions. S2L-PO uses a smaller model to generate part of each rollout group early in training and progressively anneals to fully on-policy GRPO.

and regresses to significantly lower Pass@1 in later stages, our policy perturbation proves to be more stable, converges faster, and yields superior results. In Fig. 4 and Table 1, we further observe that our method consistently reaches a higher performance ceiling than standard GRPO. For example, in the Qwen3-8B-Base setting, using a 1.7B explorer improves performance by approximately 9% compared to the baseline. The initial boost from the smaller model's diversity builds a stronger foundation, allowing the larger model to stabilize at this superior level. Notably, this improvement comes with reduced computational overhead. By offloading a portion of rollout generation to a smaller model and allowing for the reuse of these off-policy trajectories, we significantly reduce the total training FLOPs. As shown in Table 1, S2L-PO achieves consistent improvements across two model families (Qwen3 and InternLM2.5) on four benchmarks with varying scale configurations. As illustrated in Table 2, our method outperforms standard GRPO on CommonsenseQA (OOD). Specifically, for Qwen3-8B-Base, the S2L-PO-4B variant achieves an accuracy of 67.8% compared to 63.9% for the vanilla baseline, indicating that the diverse exploration preserves the model's general reasoning capabilities and improves robustness. We further extended this to Qwen3-14B-Base using S2L-PO-4B, observing similar gains over the vanilla GRPO baseline.

### 4.3. Diversity Analysis

**Quantitative measurement of policy-level diversity.** To validate that S2L-PO's gains stem from policy-level diversity, we design three complementary metrics measured on

*Table 1.* Cross-family main results (Pass@1, %). We evaluate S2L-PO across two model families (Qwen3 and InternLM2.5) on four benchmarks. $\Delta$ denotes improvement over the GRPO baseline.

| Family | Method | AIME24 | AIME25 | MATH-500 | OlympiadBench |
|---|---|---|---|---|---|
| Qwen3 | 8B GRPO | 15.0 | 12.1 | 57.3 | 18.1 |
| | **1.7B→8B S2L** | **23.8** | **22.5** | **61.5** | **19.7** |
| | $\Delta$ | +8.8 | +10.4 | +4.2 | +1.7 |
| | 14B GRPO | 18.0 | 12.9 | 58.7 | 18.9 |
| | **4B→14B S2L** | **24.4** | **14.6** | **62.7** | **21.9** |
| | $\Delta$ | +6.4 | +1.7 | +4.0 | +3.0 |
| InternLM 2.5 | 7B GRPO | 0.1 | 0.1 | 18.6 | 2.2 |
| | **1.8B→7B S2L** | **4.6** | **3.5** | **22.6** | **3.4** |
| | $\Delta$ | +4.5 | +3.4 | +4.0 | +1.2 |

*Table 2.* Out-of-domain evaluation on CommonsenseQA. Accuracy (%) of strong models trained on math data and evaluated on CommonsenseQA without additional tuning.

| | Qwen3-8B-Base | | | Qwen3-14B-Base | |
|---|---|---|---|---|---|
| | GRPO | S2L-PO-1.7B | S2L-PO-4B | GRPO | S2L-PO-4B |
| CommonsenseQA | 63.9 | 64.2 | **67.8** | 67.2 | **70.7** |

AIME24 with $K = 64$ rollouts: Self-BLEU (to reflect text repetition), Edit Diversity (to reflect token-level difference), and Unique Answer Ratio (to reflect proportion of distinct final answers). As shown in Table 3, all three metrics are monotonic with model size. The 1.7B model achieves 21% higher Unique Answer Ratio than 14B (0.576 vs. 0.476), confirming genuine strategy-level diversity.

**Controlled experiment on rollout diversity.** We filter out diverse rollouts from the small model so its diversity metrics match the large model. As shown in Table 4, performance drops back to the GRPO baseline, demonstrating that S2L-PO's gains are driven by the small model's policy-level diversity, not by other factors such as off-policy mixing.

### 4.4. Ablation Study

**Pure small-model rollouts are not sufficient for sustained performance gains.** Given the superior exploration capability of small models demonstrated above, a natural question arises: can we rely *exclusively* on small-model rollouts throughout training? Fig. 5 addresses this by evaluating a "small-only" baseline (Chen et al., 2026; Wang et al., 2025b) that never transitions to the standard GRPO. Initially, this baseline exhibits rapid performance gains, outpacing the vanilla GRPO. However, this advantage is transient: as training progresses, performance plateaus and eventually regresses, failing to reach the peak performance achieved by our progressive annealing method. We attribute this to the *widening distribution shift* between the static small-model explorer and the evolving large-model learner.

**Progressive transition vs. abrupt switch: gradual handover is strictly better.** Having established the need for a transition, we further investigate how this handover should

*Table 3.* Diversity metrics across model scales on AIME24 ($K = 64$). All metrics are strictly monotonic: smaller models are more diverse.

| Model | Self-BLEU ↓ | Edit Div. ↑ | Unique Ans. ↑ |
|-------|-------------|-------------|----------------|
| 1.7B | **0.314** | **0.788** | **0.576** |
| 4B | 0.334 | 0.773 | 0.523 |
| 8B | 0.336 | 0.769 | 0.492 |
| 14B | 0.352 | 0.760 | 0.476 |

*Table 4.* Controlled experiment: removing diversity from the small model's rollouts eliminates S2L-PO's advantage.

| Config | AIME24 | AIME25 |
|--------|--------|--------|
| S2L-PO (1.7B→8B) | **23.8** | **22.5** |
| S2L-PO (w/o diversity) | 14.7 | 12.0 |
| GRPO Baseline | 15.0 | 12.1 |

be executed. We compare our *progressive* annealing (linear decay) against an *abrupt* two-phase switch. Fig. 6 shows that the progressive transition consistently outperforms the abrupt switch. The abrupt strategy introduces a sharp shock to the training distribution, causing instability as the model struggles to adapt to the sudden loss of external guidance. In contrast, a gradual annealing allows the larger model to smoothly absorb the exploration benefits and progressively adapt its own policy to the high-quality regions discovered by the explorer, avoiding optimization divergence.

**Ablation on transition length: insufficient annealing degrades performance.** Finally, we analyze the impact of the transition duration. Fig. 7 compares schedules with different annealing lengths (*e.g.*, transitioning over 8 steps vs. 5 steps). Results indicate that shortening the transition phase leads to worse training stability and a lower performance ceiling. This suggests that the handover from small-model-assisted rollouts to predominantly larger-model rollouts is a meaningful control knob: if the transition is too fast, the larger model may not have sufficient time to digest the diverse exploration signals before being forced back to its own limited distribution. Therefore, a sufficiently long annealing period is essential for maximizing the downstream gains of the proposed method.

## 5. Related Work

### 5.1. The Evolution of RLVR

Reinforcement learning is a key paradigm for aligning large language models (LLMs) and improving reasoning ability in post-training, and recent practice is gradually shifting from preference-centric RLHF to RL with verifiable rewards (RLVR) that leverages automatically checkable signals (Guo et al., 2025; Kaufmann et al., 2023; Zhao et al., 2025). Early RLHF systems commonly relied on PPO (Schulman et al., 2017) as an online optimizer, but PPO-style training typi-

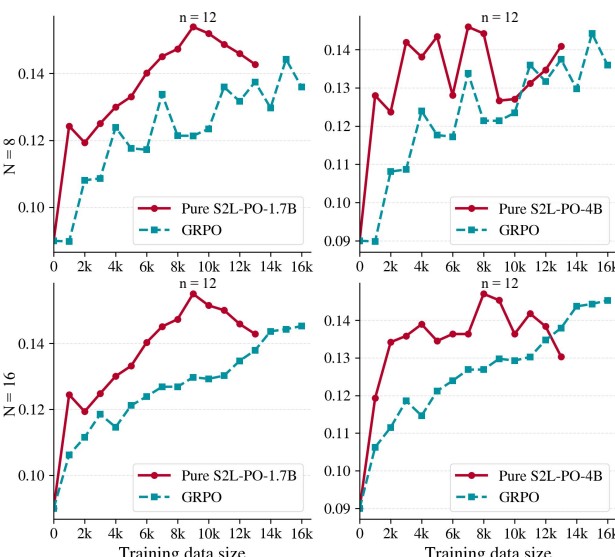

*Figure 5.* Pure small-model rollouts are insufficient for sustained improvement. Here $N$ denotes the number of GRPO rollouts and $n$ denotes the number of small-model rollouts, allowing to match total compute across settings.

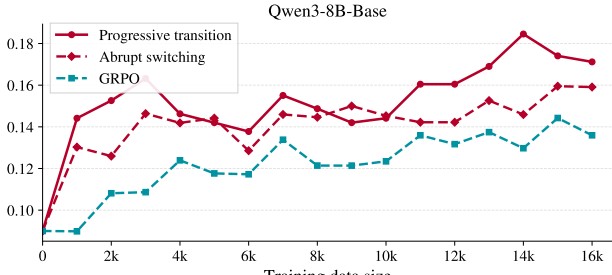

*Figure 6.* Progressive transition vs. abrupt two-phase switching.

cally requires expensive on-policy rollouts and maintaining multiple synchronized components (*e.g.*, policy, reference model, and often a critic), leading to considerable engineering complexity and computational overhead. To simplify training, Direct Preference Optimization (DPO) (Rafailov et al., 2023) rewrites KL-regularized preference learning into a closed-form classification objective, avoiding online rollouts and an explicit critic, and thus substantially streamlining the pipeline. More recently, Group Relative Policy Optimization (GRPO) (Shao et al., 2024) replaces the critic with group-relative advantage estimation using within-group statistics, reducing training cost while retaining PPO-style update stability, and is a standard baseline for reasoning-oriented RL post-training. Nevertheless, RL post-training for reasoning can still be dominated by rollout cost and limited sample efficiency (Gao et al., 2025; Hassani et al., 2025; Lanchantin et al., 2025; Mroueh et al., 2025; Wang et al., 2025b; Yu et al., 2025; Zhang et al., 2025a; Zheng et al., 2025), especially for long-horizon reasoning tasks that require repeated sampling, scoring, and backpropagation over long sequences.

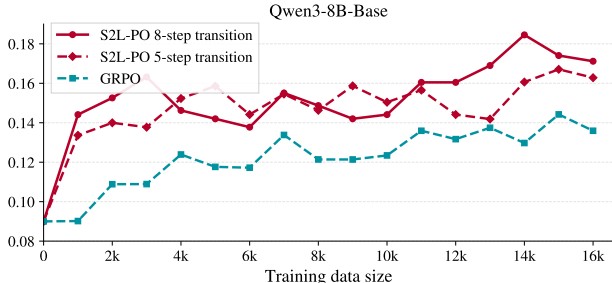

*Figure 7.* Ablation on transition length. We compare progressive annealing schedules that reduce the small-model rollout ratio to zero over the first 8 steps versus the first 5 steps.

### 5.2. Diversity and Exploration in GRPO-Style Training

A central practical factor for GRPO-style methods is the diversity of candidate trajectories sampled for each prompt: when the sampled group becomes overly homogeneous or degenerates, advantage estimation and gradient signals can deteriorate, potentially causing entropy collapse, mode collapse, and insufficient exploration (Hao et al., 2025; Jin et al., 2025; Yu et al., 2025). Most existing approaches encourage exploration by injecting randomness at the *token level*, *e.g.*, via higher temperature, top-$p$ sampling, or entropy regularization (Huang et al., 2025; Lin et al., 2024; Nguyen et al., 2024; Shi et al., 2024a; Wang et al., 2025c; Yang et al., 2025b;c; Zhuang et al., 2025). However, such action-space stochasticity is local and step-wise, and may not reliably yield *trajectory-level* structured diversity; moreover, aggressively increasing token uncertainty can hurt solution quality and training stability (Wang et al., 2025a). Beyond decoding-time randomness, several works improve group diversity through data- or objective-level interventions, such as selecting more diverse response types or explicitly rewarding within-group diversity (Anschel et al., 2025; Bamba et al., 2025; Chen et al., 2025; Zhang & Zuo, 2025; Zhang et al., 2025c). While effective in some settings, these approaches often require additional engineering or computation, and their gains may be less robust when transferring to new tasks or distributions.

Compared with token-level randomness and dataset-level heuristics, exploration via *policy*-level perturbation has received relatively less attention in LLM RL post-training. Recent off-policy methods (Chen et al., 2026; Lanchantin et al., 2025; Wang et al., 2025b) reuse previously generated rollouts or leverage external data to reduce sampling cost, but purely offline rollouts struggle to sustain performance improvement as the learner's policy evolves, due to widening distribution shift. Ensemble-based approaches can provide diverse policies but require maintaining multiple models of comparable capacity, incurring significant additional cost. S2L-PO instead introduces policy-level diversity at near-zero cost by reusing an existing smaller model from the same family, and its progressive annealing

strategy ensures sustained improvement by smoothly transitioning from small-model exploration to on-policy learning, avoiding the performance plateau inherent in purely offline approaches.

## 6. Conclusion

We have presented S2L-PO, a new framework that enhances GRPO by utilizing smaller models as structured explorers for larger learners. Because smaller models obtained via parameter-level compression (*e.g.*, distillation) inherently exhibit policy-level diversity, we provide empirical and theoretical evidence that adding this diversity to standard GRPO leads to more coherent exploration and improved learning signals than injecting token-level randomness alone. With a designed annealing strategy to balance exploration and exploitation, S2L-PO achieves significant gains in mathematical reasoning tasks while reducing rollout compute and accelerating convergence. Our results demonstrate that leveraging the inherent diversity from parameter-level perturbation is a powerful and efficient strategy for RL training.

## Impact Statement

This work exclusively relies on publicly available open-source datasets that have been widely used and validated in prior academic research. No new text, images, audio, or video content is generated or collected as part of this study. All datasets are used strictly for research purposes, and we do not engage in any commercial deployment or application of the data or the trained models.

## Acknowledgements

This work was partly supported by the National Natural Science Foundation of China (Grant No. 62576191), the Shenzhen Science and Technology Program (ZDCY20250901103533010) and Tsinghua SIGS KA Cooperation Fund.

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

# A. Reproducibility Statement

To facilitate reproducibility and transparency, we will release the complete codebase of this project as open-source software. The overall methodology and algorithmic design are described in detail in Section 3. The experimental setup is specified in Section 4.1, and the complete training protocols, implementation details, and key hyperparameter configurations are provided in the appendix. These materials together are sufficient to reproduce all experimental results reported in this paper.

# B. Use of Large Language Models

During the preparation of this manuscript, we used a large language model solely for language editing purposes, including improving grammar, clarity, and overall readability at the sentence and paragraph levels. The model was not used to generate research ideas, design methods, conduct experiments, analyze results, or draw scientific conclusions. All technical content, experimental design, analyses, and interpretations were written, verified, and approved by the authors. Every model-assisted edit was carefully reviewed to ensure correctness, and the authors take full responsibility for the accuracy and integrity of the final manuscript.

# C. Hyperparameter Settings

All experiments are trained with GRPO using a fixed set of core hyperparameters across runs. We set the training batch size to 1024, the maximum prompt length to 512 tokens, and the maximum response length to 4096 tokens, while filtering overlong prompts and treating truncation as an error to avoid silent data corruption. The actor policy is optimized with a learning rate of $1 \times 10^{-6}$ using PPO-style updates with a mini-batch size of 16 and a per-GPU micro-batch size of 2. We enable KL regularization between the actor and a reference policy with KL coefficient $1 \times 10^{-3}$ and use a low-variance KL estimator, while setting the entropy bonus coefficient to 0 and not incorporating KL into the reward. For rollout and log-probability computation, we use micro-batching with size 2 per GPU, and keep tensor model parallelism at 1. All runs are performed on 8 GPUs, with checkpointing enabled at every training step and evaluation triggered every 100 steps, and training is terminated by a fixed number of training steps rather than a fixed number of epochs. We use a progressive off-policy to on-policy schedule over 16 logical steps, where the first 8 logical steps linearly decrease the offline sampling ratio from 1 to 0 and increase the online rollout ratio from 0 to 1, and the remaining logical steps are fully on-policy.

# D. Deployment of Progressive Off-to-On GRPO

In this appendix, we describe how our progressive off-policy to on-policy schedule is deployed within the GRPO training pipeline. The key idea is to generate candidate trajectories from a mixture of an offline source and the current policy, and to linearly anneal the offline contribution to zero. This staged procedure provides low-cost exploration early in training while ensuring that the final policy is optimized under its own on-policy distribution.

# E. Formal Proofs for Theoretical Analysis

In this appendix we provide rigorous proofs for the theoretical claims in Section 3.1. We formalize the distinction between token-level and policy-level perturbations in terms of cross-step covariance bounds for the GRPO gradient signal.

## E.1. Notation and Setup

Let $o = (a_1, \ldots, a_L)$ denote a sampled trajectory for prompt $q$, and $o^\star = (a_1^\star, \ldots, a_L^\star)$ a deterministic reference decoding trace under the base policy $\pi_\theta$. Define the prefix match indicator $M_t := \mathbb{I}\{(a_1, \ldots, a_t) = (a_1^\star, \ldots, a_t^\star)\}$. For the per-step score function $\mathbf{u}_{i,t} := \nabla_\theta \log \pi_\theta(a_{i,t} \mid s_{i,t})$, define the scalar projection $z_{i,t} := \langle \mathbf{u}_{i,t}, \mathbf{v} \rangle$ for a unit vector $\mathbf{v}$, and assume $|z_{i,t}| \leq B$.

**Lemma E.1** (Prefix Match Decay). *Under token-level perturbation with per-step deviation probability lower-bounded by $p > 0$, the prefix match probability satisfies:*

$$\Pr(M_t = 1) = \prod_{j=1}^{t} \Pr(a_j = a_j^\star \mid M_{j-1} = 1) \leq (1-p)^t.$$

*Proof.* Since $\{M_t = 1\} = \{M_{t-1} = 1\} \cap \{a_t = a_t^\star\}$, the chain rule gives $\Pr(M_t = 1) = \prod_{j=1}^t \Pr(a_j = a_j^\star \mid M_{j-1} = 1)$, with base case $\Pr(M_0 = 1) = 1$. The assumption $\Pr(a_j \neq a_j^\star \mid M_{j-1} = 1) \geq p$ yields each factor $\leq 1 - p$, so the product $\leq (1 - p)^t$. $\qquad\square$

**Proposition E.2** (Token-Level Covariance Upper Bound). *Let $f_t, g_s$ be random variables measurable with respect to the trajectory prefix $(a_1, \ldots, a_t)$ and $(a_1, \ldots, a_s)$ respectively, with $\|f_t\|_\infty, \|g_s\|_\infty \leq 1$ and $t < s$. Then:*

$$|\mathrm{Cov}(f_t, g_s \mid q)| \leq 5 \Pr(M_t = 0).$$

*Proof.* We apply the law of total covariance with conditioning variable $M_t \in \{0, 1\}$:

$$\mathrm{Cov}(f_t, g_s) = \underbrace{\mathbb{E}[\mathrm{Cov}(f_t, g_s \mid M_t)]}_{(I)} + \underbrace{\mathrm{Cov}(\mathbb{E}[f_t \mid M_t], \mathbb{E}[g_s \mid M_t])}_{(II)}.$$

**Term (I):** When $M_t = 1$, the prefix $(a_1, \ldots, a_t)$ is fixed to $(a_1^\star, \ldots, a_t^\star)$, so $f_t$ is a constant and $\mathrm{Cov}(f_t, g_s \mid M_t = 1) = 0$. When $M_t = 0$, Cauchy–Schwarz gives $|\mathrm{Cov}(f_t, g_s \mid M_t = 0)| \leq 1$. Thus $|(I)| \leq \Pr(M_t = 0) \cdot 1$.

**Term (II):** Let $\alpha = \Pr(M_t = 1)$, $\mu_k = \mathbb{E}[f_t \mid M_t = k]$, $\nu_k = \mathbb{E}[g_s \mid M_t = k]$ for $k \in \{0, 1\}$. Since $\mathbb{E}[f_t \mid M_t]$ and $\mathbb{E}[g_s \mid M_t]$ are functions of a Bernoulli variable:

$$(II) = \alpha(1 - \alpha)(\mu_1 - \mu_0)(\nu_1 - \nu_0).$$

Using $\alpha(1 - \alpha) \leq 1 - \alpha = \Pr(M_t = 0)$ and $|\mu_1 - \mu_0|, |\nu_1 - \nu_0| \leq 2$:

$$|(II)| \leq 4 \Pr(M_t = 0).$$

Combining: $|\mathrm{Cov}(f_t, g_s)| \leq \Pr(M_t = 0) + 4 \Pr(M_t = 0) = 5 \Pr(M_t = 0)$. $\qquad\square$

**Corollary E.3** (Gradient Projection Covariance). *For the scalar projections $z_{i,t} = \langle \mathbf{u}_{i,t}, \mathbf{v} \rangle$ with $|z_{i,t}| \leq B$ and $t < s$:*

$$|\mathrm{Cov}(z_{i,t}, z_{i,s} \mid q)| \leq 5B^2 \Pr(M_t = 0).$$

*Proof.* Apply Proposition E.2 to $f_t = z_{i,t}/B$ and $g_s = z_{i,s}/B$, then scale by $B^2$. $\qquad\square$

**Proposition E.4** (Policy-Level Covariance Lower Bound). *Let $\tilde{z}_{i,t} = z_{i,t} + \mathbf{v}^\top H_t \delta_\theta$ denote the first-order perturbed score projection, where $H_t = \nabla_\theta^2 \log \pi_\theta(a_t \mid s_t)$ and $\delta_\theta$ has zero mean and covariance $\Sigma_\delta$. If $\gamma := \mathbb{E}[\mathbf{v}^\top H_t \Sigma_\delta H_s^\top \mathbf{v} \mid q] > 0$, then for $t < s$:*

$$|\mathrm{Cov}(\tilde{z}_{i,t}, \tilde{z}_{i,s} \mid q)| \geq \gamma - 5B^2 \overset{\mathrm{std}}{\Pr}(M_t = 0) - O(\|\delta_\theta\|^3),$$

*where $\Pr^{\mathrm{std}}$ denotes prefix divergence under the standard (unperturbed) temperature.*

*Proof.* Expanding by bilinearity of covariance:

$$\mathrm{Cov}(\tilde{z}_t, \tilde{z}_s) = \mathrm{Cov}(z_t, z_s) + \mathrm{Cov}(z_t, \mathbf{v}^\top H_s \delta_\theta) + \mathrm{Cov}(\mathbf{v}^\top H_t \delta_\theta, z_s) + \mathrm{Cov}(\mathbf{v}^\top H_t \delta_\theta, \mathbf{v}^\top H_s \delta_\theta).$$

**Cross-terms vanish:** In the zero-order approximation, $z_t$ and $H_s$ are computed along the base policy trajectory and are independent of $\delta_\theta$. Since $\mathbb{E}[\delta_\theta] = 0$, both $\mathbb{E}[z_t \cdot \mathbf{v}^\top H_s \delta_\theta]$ and $\mathbb{E}[z_t] \cdot \mathbb{E}[\mathbf{v}^\top H_s \delta_\theta]$ vanish, giving $\mathrm{Cov}(z_t, \mathbf{v}^\top H_s \delta_\theta) = 0$. The trajectory's $O(\|\delta_\theta\|)$ dependence on $\delta_\theta$ contributes $O(\|\delta_\theta\|^3)$ to the covariance.

**Perturbation term:** Both means vanish ($\mathbb{E}[\delta_\theta] = 0$), so:

$$\mathrm{Cov}(\mathbf{v}^\top H_t \delta_\theta, \mathbf{v}^\top H_s \delta_\theta) = \mathbb{E}[(\mathbf{v}^\top H_t \delta_\theta)(\delta_\theta^\top H_s^\top \mathbf{v})] = \mathbb{E}[\mathbf{v}^\top H_t \Sigma_\delta H_s^\top \mathbf{v} \mid q] = \gamma,$$

where we used the independence of $\delta_\theta$ and the trajectory (at zero order) to factor the expectation.

**Combining:** $\mathrm{Cov}(\tilde{z}_t, \tilde{z}_s) = \mathrm{Cov}(z_t, z_s) + \gamma + O(\|\delta_\theta\|^3)$. By the reverse triangle inequality and Corollary E.3:

$$|\mathrm{Cov}(\tilde{z}_t, \tilde{z}_s)| \geq \gamma - |\mathrm{Cov}(z_t, z_s)| - O(\|\delta_\theta\|^3) \geq \gamma - 5B^2 \overset{\mathrm{std}}{\Pr}(M_t = 0) - O(\|\delta_\theta\|^3).$$

The lower bound is positive when $\gamma$ dominates, *i.e.*, when Hessian alignment is strong and the standard-temperature prefix divergence is moderate. $\qquad\square$

*Remark* E.5. The Hessian alignment condition $\gamma > 0$ is natural for same-family distilled models: since $\delta_\theta$ arises from structured capacity reduction that affects shared feature-extraction layers, the Hessians $H_t$ and $H_s$ project $\delta_\theta$ onto similar directions across different decoding steps, yielding consistently positive alignment. This contrasts sharply with token-level perturbation, where the covariance upper bound in Corollary E.3 becomes vacuous as $\Pr(M_t = 0) \to 1$ with increasing temperature.

*Remark* E.6 (Fixed vs. random $\delta_\theta$). In practice, $\delta_\theta = \theta_{\text{small}} - \theta_{\text{large}}$ is a fixed vector determined by the specific small model. In the theoretical analysis above, we model $\delta_\theta$ as a zero-mean random variable (representing the uncertainty across possible distillation outcomes) in order to derive the expectation-based lower bound. For a fixed $\delta_\theta$, the covariance expansion still holds: the fourth term becomes $\text{Cov}(\mathbf{v}^\top H_t \delta_\theta, \mathbf{v}^\top H_s \delta_\theta \mid q)$, which is the covariance of Hessian projections over trajectory randomness. The lower bound then depends on the specific alignment of this fixed $\delta_\theta$ with the Hessian structure, rather than on the averaged alignment $\gamma$.

*Remark* E.7 (Bound on $B$ under softmax parameterization). Under softmax parameterization, $\log \pi_\theta(a \mid s) = \ell_a - \log \sum_{a'} \exp(\ell_{a'})$, where $\ell_a$ denotes the logit for action $a$. The gradient with respect to the logit parameters is $\nabla_\ell \log \pi_\theta(a \mid s) = e_a - \pi_\theta(\cdot \mid s)$, where $e_a$ is a one-hot vector. Each component has absolute value at most 1, so $\|\nabla_\ell \log \pi_\theta(a \mid s)\|_2 \leq \sqrt{|\mathcal{V}|}$ where $|\mathcal{V}|$ is the vocabulary size. Consequently, for the scalar projection $z_{i,t} = \langle \mathbf{u}_{i,t}, \mathbf{v} \rangle$ with $\|\mathbf{v}\| = 1$, one can take $B = \sqrt{|\mathcal{V}|}$ in the logit-parameter setting. For the full model parameters $\theta$ (beyond the last layer), gradient clipping during training provides an effective bound on $B$.

### E.2. Summary: Token-Level vs. Policy-Level Signal Growth

The key qualitative difference between the two perturbation mechanisms can be summarized as follows:

|  | Token-level (high temp.) | Policy-level (param. perturb.) |
| --- | --- | --- |
| Covariance | Upper bound: $\leq 5B^2 \Pr^{\text{tok}}(M_t{=}0)$ | Lower bound: $\geq \gamma - 5B^2 \Pr^{\text{std}}(M_t{=}0) - O(\|\delta_\theta\|^3)$ |
| Behavior | $\Pr^{\text{tok}}(M_t{=}0) \to 1$ fast $\Rightarrow$ vacuous | $\gamma > 0$ independent of $t \Rightarrow$ positive |
| Signal growth | $\mathbb{E}[\|\sum_t \mathbf{u}_t\|^2] \sim O(L)$ (random walk) | $\mathbb{E}[\|\sum_t \tilde{\mathbf{u}}_t\|^2] \gtrsim O(L^2)$ (constructive) |

Policy-level perturbation injects a shared, time-invariant signal $\mathbf{v}^\top H_t \delta_\theta$ into each step's score function. This common component induces positive cross-step covariance, causing gradient contributions to reinforce constructively across the horizon rather than cancelling like a random walk. This is the theoretical basis for why smaller models, as structured policy perturbations, provide more informative exploration signals for GRPO training.

## F. Limitations

Our empirical evaluation is constrained by computational resources, preventing exhaustive coverage of all prominent model families and benchmark categories. In particular, we have not validated S2L-PO on tasks beyond mathematical reasoning that rely on non-verifiable or open-ended rewards. The capability boundary of S2L-PO under broader model scales, task domains, and modalities remains to be explored in future work.

