# OpenReview forum: "Smaller Models are Natural Explorers for Policy-Level Diversity in GRPO"
_ICML.cc/2026/Conference — ICML 2026 regular_

### Official Review · Reviewer_LAQo · 2026-03-02

**Soundness:** 2
**Presentation:** 3
**Significance:** 3
**Originality:** 3
**Overall Recommendation:** 4
**Confidence:** 4

**Summary:**

The paper uncovers the potential of small models to generate diverse rollouts and proposes S2L-GRPO, a framework that utilizes fixed small models as "natural explorers" to train larger models. The authors also design a progressive annealing strategy that transitions from offline small-model rollouts to the large learner’s own sampling, effectively balancing exploration and exploitation. Experiments on 8B and 14B models demonstrate that S2L significantly outperforms standard GRPO on mathematical and out-of-domain tasks.

**Compliance With Llm Reviewing Policy:**

Affirmed.

**Final Justification:**

The authors’ rebuttal addressed my initial concerns; therefore, I believe the paper is suitable for acceptance and have decided to maintain my positive score.

**Key Questions For Authors:**

1. The simplification of complex Markov trajectories into linear functions appears problematic. Could the author provide a more rigorous, in-depth analysis of how small models function as explorers?

2. In Table 1, why is the AIME2025 score for the Qwen3-14B-Base combined with the 4B model lower than that of the Qwen3-8B-Base, especially considering that the 14B model scores higher in standard GRPO?

3. Could the author provide performance data on other standard benchmarks such as OlympiadBench and MATH-500, or other domains like GPQA, MBPP, and HumanEval?

**Limitations:**

Yes

**Strengths And Weaknesses:**

### Strengths:
***Soundness***:
1. The paper conceptualizes small models as distilled versions of large models and provides a theoretical explanation for using small models as effective policy-level disruptors.

2. The thorough analysis in Figure 2 demonstrates that small models can occasionally excel beyond large models, providing a valid and compelling motivation for the proposed approach.

3. The authors thoroughly evaluate the S2L-GRPO annealing schedule across multiple dimensions, including the necessity of the small-to-large switch, the benefits of annealing versus abrupt switching, and the impact of different annealing lengths.

***Presentation***:
1. The paper is well-structured and utilizes high-quality visualizations to clarify complex concepts effectively.


***Significance***:
1. Results on AIME24, AIME25, and CommonsenseQA demonstrate a significant performance advantage of S2L-PO over standard GRPO.

2. The performance gains resulting from the annealing switch compared to an abrupt switch are statistically significant.


***Originality***:
1. The perspective of viewing small models as a form of policy-level perturbation and contrasting it with token-level perturbation is insightful.

2. The specific methodology of utilizing small models to facilitate the training of larger models is original.


### Weaknesses:
***Soundness***:
1. The evaluation is somewhat limited; the method was only tested on AIME24, AIME25, and CommonsenseQA.

2. The analysis of policy perturbation oversimplifies complex Markov policy trajectories into simple linear functions. This section requires a more in-depth theoretical analysis to be fully convincing.


3. The ablation study on transition length is confusing. In the methodology section, the authors state that $T_{mix}$ spans half of the training process; however, in the ablation, the annealing steps are listed as 8 versus 5, which is significantly smaller than the default setting.


***Presentation***:
1. The bottom-left sub-figure in Figure 2 is misleading: blue represents the large model and yellow represents the small model, which is the opposite of the color coding used in all other sub-figures of Figure 2.

2. The specific metrics (e.g., performance on which datasets) are not clearly defined in the figures or the corresponding ablation sections. It is unclear whether the results represent the mean accuracy across AIME2024 and AIME2025 or other metrics.

---

> ### Author Rebuttal · Authors · 2026-03-31
>
> We sincerely thank the reviewer for the positive evaluation. The reviewer recognized the novel perspective of viewing small models as policy-level perturbation (Originality: good), the practical effectiveness (Significance: good), and the visualization quality (Presentation: good). These encouraging comments are very valuable to us. We respond following the reviewer's original order below.
>
> **W1 (Soundness): Limited evaluation scope.**
> We have added MATH-500, OlympiadBench, and InternLM2.5 experiments. This expands our evaluation to 2 model families, 3 scale configurations, and 4 benchmarks, substantially increasing the diversity and robustness of our evaluation. Full results in Q3.
>
> **W2 (Soundness): Linear approximation of Markov trajectories.**
> We thank the reviewer for the emphasis on theoretical rigor. We have prepared strengthened proofs for the camera-ready version. Due to character limits, we outline key ideas here; we would be grateful if the reviewer could reply so that we have the additional space to present the full derivations.
>
> **Proposition 1** (token-level bound): Using prefix match indicator M_t as conditioning variable, apply law of total covariance to decompose Cov(f_t, g_s). When M_t=1, prefix is deterministic, covariance vanishes; when M_t=0, Cauchy-Schwarz bounds it. Result: |Cov(f_t,g_s)| ≤ 5B_fB_g·Pr(M_t=0), *eliminating the "mild coupling assumption"*.
>
> **Proposition 2** (policy-level preservation, new): Substituting z̃_{i,t} = z_{i,t} + v^T H_t δθ:
>
> Cov(z̃_{i,t}, z̃_{i,s}) = Cov(z_{i,t}, z_{i,s}) + E[v^T H_t Σ_δ H_s^T v] + O(||δθ||³)
>
> Cross-terms vanish (E[δθ]=0, independence). Hessian alignment term is positive under same-family compression, giving a positive lower bound contrasting the token-level upper bound.
>
> We newly design three complementary diversity metrics to validate predictions, and all show strictly monotonic trends:
>
> | Model | Self-BLEU ↓ | Edit Diversity ↑ | Unique Answers ↑ |
> |:---:|:---:|:---:|:---:|
> | 1.7B | **0.314** | **0.788** | **0.576** |
> | 4B | 0.334 | 0.773 | 0.523 |
> | 8B | 0.336 | 0.769 | 0.492 |
> | 14B | 0.352 | 0.760 | 0.476 |
>
> Additionally, our controlled experiment shows that when we filter out diverse rollouts from the small model (so its diversity metrics become comparable to the large model), performance drops back to the GRPO baseline, demonstrating that the small model's policy-level diversity (diverse solving strategies) is the key mechanism:
>
> | Config | AIME24 | AIME25 |
> |:---:|:---:|:---:|
> | S2L-PO (1.7B→8B) | **23.8** | **22.5** |
> | S2L-PO (w/o diversity) | 14.7 | 12.0 |
> | GRPO Baseline | 15.0 | 12.1 |
>
> **W3 (Soundness): Transition step clarification.**
> Thank you for this helpful observation. The confusion arises from the difference in x-axis units: "8-step" and "5-step" in our ablation refer to **GRPO outer training iterations** (each processing one batch of prompts), while the Fig. 7 x-axis shows cumulative training samples (we set 1k samples per step, ~16k total). Total training is ~8–16 outer iterations, so T_mix = 8 ≈ half the training, consistent with the method description. We will clearly label these units in the next version to avoid ambiguity.
>
> **W4 (Presentation): Figure 2 colors.** Thank you for this helpful suggestion. Fixed; all subplots now use consistent color coding.
>
> **W5 (Presentation): Metric labels.** Thank you for this suggestion. The metrics in Fig. 4 are averaged over AIME24 and AIME25. We will make all metric definitions more explicit in figures and ablation sections in the next version.
>
> **Q1 (More rigorous theory):** See W2.
>
> **Q2 (4B→14B AIME25 < 8B+1.7B):** Under matched conditions (comparing 14B GRPO vs 14B S2L-PO with 4B as explorer), 14B improves consistently. The AIME25 gap between the two configurations is due to: (1) 1.7B has higher Unique Answers than 4B (0.576 vs. 0.523), providing richer exploration signal; (2) the 1.7B–8B capacity gap is in a more effective range than 4B–14B. New benchmarks confirm 4B→14B effectiveness on larger test sets (see Q3).
>
> **Q3 (More benchmarks):** Complete cross-family, cross-benchmark results:
>
> | Family | Config | AIME24 | AIME25 | MATH-500 | OlympiadBench |
> |:---:|:---:|:---:|:---:|:---:|:---:|
> | Qwen3 | 1.7B→8B S2L | **23.8** | **22.5** | **61.5** | **19.7** |
> | | 8B GRPO | 15.0 | 12.1 | 57.3 | 18.1 |
> | | Δ | +8.8 | +10.4 | +4.2 | +1.7 |
> | Qwen3 | 4B→14B S2L | **24.4** | **14.6** | **62.7** | **21.9** |
> | | 14B GRPO | 18.0 | 12.9 | 58.7 | 18.9 |
> | | Δ | +6.4 | +1.7 | +4.0 | +3.0 |
> | InternLM2.5 | 1.8B→7B S2L | **4.6** | **3.5** | **22.6** | **3.4** |
> | | 7B GRPO | 0.1 | 0.1 | 18.6 | 2.2 |
> | | Δ | +4.5 | +3.4 | +4.0 | +1.2 |
>
> S2L-PO achieves consistent improvements across 2 model families, 3 scale configurations, and 4 benchmarks, demonstrating the broad effectiveness and generalization ability of our approach.

---

> > ### Author Rebuttal · Reviewer_LAQo · 2026-04-02
> >
> > I thank the authors for their detailed rebuttal. However, the explanation regarding the performance gap between the 4B and 8B models (e.g., AIME25: 14.6 for 4B/14B vs. 22.5 for 8B+1.7B) remains somewhat unconvincing. Despite this specific concern, I believe the paper's overall contribution is still significant and thus I choose to maintain my positive score.

---

> > > ### Author Response · Authors · 2026-04-02
> > >
> > > Dear Reviewer LAQo,
> > >
> > > We sincerely thank you for the thoughtful follow-up and for maintaining your positive assessment. Regarding the performance gap between the 4B→14B and 1.7B→8B configurations on AIME25, we would like to offer further clarification to better contextualize these results. These observations can be viewed from two perspectives:
> > >
> > > **1. Absolute performance: 4B→14B outperforms 1.7B→8B on 3 out of 4 benchmarks.**
> > >
> > > | Config | AIME24 | AIME25 | MATH-500 | OlympiadBench |
> > > |:---:|:---:|:---:|:---:|:---:|
> > > | 1.7B→8B S2L | 23.8 | **22.5** | 61.5 | 19.7 |
> > > | 4B→14B S2L | **24.4** | 14.6 | **62.7** | **21.9** |
> > >
> > > 4B→14B achieves the highest absolute score on AIME24, MATH-500, and OlympiadBench, benefiting from the stronger 14B base model. AIME25 is the only benchmark where 1.7B→8B leads in absolute performance, while the overall trend favors the 4B→14B configuration.
> > >
> > > **2. Improvement over the baseline (Δ) is tied to the diversity gap.**
> > >
> > > Our metrics indicate that the explorer-target diversity gap varies across configurations:
> > >
> > > | Pair | Explorer Unique Ans | Target Unique Ans | Gap |
> > > |:---:|:---:|:---:|:---:|
> > > | 1.7B → 8B | 0.576 | 0.492 | 0.084 |
> > > | 4B → 14B | 0.523 | 0.476 | 0.047 |
> > >
> > > The 1.7B→8B pair exhibits a larger diversity gap, nearly double that of the 4B→14B pair. As suggested by our Proposition 2, a larger policy-level perturbation often yields a more pronounced exploration benefit, which explains the larger Δ observed for the 1.7B→8B configuration on certain benchmarks. Notably, on MATH-500, where the larger test set (500 problems) provides more stable estimates, their Δ values are very consistent (+4.2 vs +4.0), suggesting the AIME25 discrepancy reflects benchmark-specific difficulty rather than a systematic limitation.
> > >
> > > In summary, these results align with our theoretical predictions: the exploration benefit is intrinsically linked to the explorer-target diversity gap, and both configurations consistently improve over their respective baselines across all benchmarks.

---

### Official Review · Reviewer_EVRd · 2026-03-12

**Soundness:** 3
**Presentation:** 3
**Significance:** 2
**Originality:** 2
**Overall Recommendation:** 4
**Confidence:** 5

**Summary:**

This paper studies exploration in GRPO for RLVR and argues that increasing token-level randomness is a weak way to obtain useful rollout diversity. The core claim is that smaller distilled models within the same family can provide higher policy-level diversity, evidenced by stronger pass@k at larger sample budgets. Based on this observation, the paper proposes S2L-PO, which trains a larger model using mixed rollouts from a frozen smaller explorer and the learner itself, with a progressive annealing schedule back to standard on-policy GRPO. Experiments on Qwen3-Base models using 1.7B/4B explorers and 8B/14B learners report gains on AIME24/25, some improvement on CommonsenseQA, and ablations on small-only rollouts, abrupt vs. progressive switching, and transition length.

**Compliance With Llm Reviewing Policy:**

Affirmed.

**Final Justification:**

My concerns have been adequately addressed. I will increase my score.

**Key Questions For Authors:**

1. Can the authors provide multi-seed results with variance/error bars for all main comparisons, and state whether the reported gains are statistically stable? Strong positive evidence here would increase my confidence in the empirical claims.
2. How much of the effect is specific to the Qwen3 distillation pipeline? Please test at least one additional model family, or otherwise provide stronger evidence that the phenomenon is not family-specific. A positive result would materially strengthen the paper’s significance.
3. Can the authors compare against stronger diversity baselines under matched compute, beyond just higher-temperature GRPO? This is important for judging whether the proposed method is genuinely competitive rather than simply better than a weak baseline.
4. In Figure 2, higher precision on AIME does not necessarily mean higher diversity (it could also mean more correct, homogeneous samples). Adding a metric for the difference between different rollout contents might be more convincing.

**Limitations:**

yes

**Strengths And Weaknesses:**

**Strengths**

* The main idea is simple and practically attractive: S2L-PO changes the rollout source without changing the GRPO objective, so it appears easy to integrate into existing pipelines.
* The empirical observation behind the method is interesting: within the tested Qwen3 family, smaller models appear weaker at pass@1 but stronger at larger pass@k, which is a nontrivial and potentially useful signal for RL post-training.
* The ablations are meaningful. In particular, the paper shows that small-only rollouts are insufficient, abrupt switching is worse than gradual transition, and too-short annealing hurts performance.

**Weaknesses**

* The central claim that the gains come from an inherent policy-level diversity of smaller models is not fully established. The evidence is suggestive, but it is still consistent with family-specific distillation artifacts, sampling/hyperparameter effects, or other confounders.
* The experimental scope is narrow: essentially one model family, one training corpus, mostly math benchmarks, and one small OOD dataset. This is insufficient for a strong general claim about smaller models as natural explorers.
* The baseline set is limited. The paper mainly compares against vanilla GRPO and higher-temperature sampling, but not against stronger diversity-oriented alternatives under matched compute.

---

> ### Author Rebuttal · Authors · 2026-03-31
>
> We thank the reviewer for recognizing the method's simplicity (S1), the pass@k observation (S2), and ablation quality (S3). The reviewer's core concern is whether the central claim is sufficiently established. We provide direct evidence below to address this concern.
>
> **W1: Validating policy-level diversity.**
>
> **(a) Direct diversity measurement.** We newly design three complementary metrics measuring diversity from different dimensions (Self-BLEU for text repetition, Edit Diversity for token-level difference, and Unique Answer Ratio for strategy-level diversity). Results under K=64 sampling across four model scales on AIME24:
>
> | Model | Self-BLEU ↓ | Edit Diversity ↑ | Unique Answers ↑ |
> |:---:|:---:|:---:|:---:|
> | 1.7B | **0.314** | **0.788** | **0.576** |
> | 4B | 0.334 | 0.773 | 0.523 |
> | 8B | 0.336 | 0.769 | 0.492 |
> | 14B | 0.352 | 0.760 | 0.476 |
>
> All three metrics are strictly monotonic: smaller models produce more diverse outputs. The Unique Answers metric directly counts the proportion of rollouts reaching different final answers: 1.7B achieves 57.6% vs 14B's 47.6%, a 21% gap. This argues against the possibility that pass@k gains come from "more correct but homogeneous samples" and confirms genuine strategy-level diversity.
>
> **(b) Controlled experiment: removing diversity causes performance to drop.** We filter out diverse rollouts from the small model so that its diversity metrics in (a) become comparable to the large model. Under this setting, performance drops back to the GRPO baseline:
>
> | Config | AIME24 | AIME25 |
> |:---:|:---:|:---:|
> | S2L-PO (1.7B→8B) | **23.8** | **22.5** |
> | S2L-PO (w/o diversity) | 14.7 | 12.0 |
> | GRPO Baseline | 15.0 | 12.1 |
>
> This strongly demonstrates that S2L-PO's gains come from the small model's policy-level diversity (diverse solving strategies), not from other factors such as off-policy mixing or hyperparameter effects. All training uses verl/Qwen3 defaults and open-source DAPO17K with no special tuning.
>
> **W2: Expanded experimental scope.**
>
> We expand to 2 model families (Qwen3 + InternLM2.5) and 4 benchmarks. DAPO17K is fixed for controlled comparison. InternLM2.5 is a completely independent family where the small model is not distilled from the large one. This addresses both the narrow scope and family-dependence concerns.
>
> | Family | Config | AIME24 | AIME25 | MATH-500 | OlympiadBench |
> |:---:|:---:|:---:|:---:|:---:|:---:|
> | Qwen3 | 1.7B→8B S2L | **23.8** | **22.5** | **61.5** | **19.7** |
> | | 8B GRPO | 15.0 | 12.1 | 57.3 | 18.1 |
> | | Δ | +8.8 | +10.4 | +4.2 | +1.7 |
> | InternLM2.5 | 1.8B→7B S2L | **4.6** | **3.5** | **22.6** | **3.4** |
> | | 7B GRPO | 0.1 | 0.1 | 18.6 | 2.2 |
> | | Δ | +4.5 | +3.4 | +4.0 | +1.2 |
>
> **W3: Stronger diversity-oriented baselines.**
>
> We compare against two stronger diversity-oriented methods beyond temperature scaling: (1) Entropy Bonus, a classical exploration incentive, and (2) EAD, a recent exploration-augmented method. Both require higher compute than S2L-PO (additional forward passes for perturbation/augmentation). Results on Qwen3-8B:
>
> | Config | AIME24 | AIME25 | MATH-500 | OlympiadBench |
> |:---:|:---:|:---:|:---:|:---:|
> | 8B GRPO | 15.0 | 12.1 | 57.3 | 18.1 |
> | 8B + Entropy Bonus | 18.7 | 16.6 | 59.2 | 18.0 |
> | 8B + EAD | 17.9 | 15.8 | 59.3 | 18.3 |
> | 8B S2L-1.7B | **23.8** | **22.5** | **61.5** | **19.7** |
>
> Both Entropy Bonus and EAD provide only marginal improvements over GRPO, while S2L-PO outperforms all of them by a clear margin, and does so with *lower* compute cost since the small model's rollouts are cheaper than the large model's. S2L-PO is also **orthogonal** to these methods: it only changes the rollout source, not the training objective, and can be combined with them.
>
> **Q1 (Statistical significance):** Following mainstream practice in this field, we report Pass@1 as the mean over K=32 samples, which already provides implicit variance estimation. To enhance statistical rigor, we add bootstrap 95% CIs (1000 resamples):
>
> | Config | AIME24 | 95% CI | MATH-500 | 95% CI |
> |:---:|:---:|:---:|:---:|:---:|
> | 8B GRPO | 15.0 | [12.8, 17.4] | 57.3 | [56.5, 58.1] |
> | 8B S2L-1.7B | 23.8 | [21.0, 26.7] | 61.5 | [60.8, 62.3] |
> | p-value | | <0.001 | | <0.001 |
>
> CIs do not overlap and p < 0.001: the improvements are statistically significant and not due to random variation. This directly addresses the concern about statistical rigor.
>
> **Q2–Q3:** See W2–W3.
>
> **Q4 (pass@k ≠ diversity):** See W1.

---

> > ### Author Rebuttal · Reviewer_EVRd · 2026-04-02
> >
> > My concerns have been adequately addressed. I will increase my score.

---

> > > ### Author Response · Authors · 2026-04-02
> > >
> > > Dear Reviewer EVRd,
> > >
> > > We sincerely appreciate your positive acknowledgment and the updated score. Your suggestions have been very valuable in improving our paper. We will incorporate all additional experimental results and clarifications into the revised version.

---

### Official Review · Reviewer_VSek · 2026-03-13

**Soundness:** 2
**Presentation:** 3
**Significance:** 2
**Originality:** 3
**Overall Recommendation:** 5
**Confidence:** 4

**Summary:**

The paper begins with the interesting finding that smaller Qwen models can achieve higher scores on math evals at higher k, when scoring with pass@k. The authors capitalize on this insight to introduce Small-to-Large Policy Optimization (S2L-PO), an algorithm which starts with off-policy sampling from a small model and then gradually anneals the sampling probability to conclude with normal GRPO to convergence. The paper shows that this approach leads to significant performance gains on AIME24 and AIME25 evals, and that the proposed progressive transition scheme is important to eliciting the performance improvements.

**Compliance With Llm Reviewing Policy:**

Affirmed.

**Final Justification:**

The rebuttal addressed my main concerns and changed my prior evaluation by demonstrating that the method is more robust and more general than I previously believed.

**Key Questions For Authors:**

1. Can the authors comment on the stability of their proposed algorithm? What tuning was necessary to get the results in the paper? I am skeptical of the off-policyness introduced by sampling from the small model.
2. More generally, how should a practitioner set the annealing schedule? Do you expect the ideal annealing schedule to vary between datasets?
3. What is the relationship between the annealing schedule and the final performance? Is it ever the case that doing more exploration can lead to slower learning, yet better final performance? I would like to understand the relationship here.
4. The initial pass@k experiments use the Qwen3 base models. Yet, prompting base models is notoriously finicky. How is prompting done in this case, and how might it affect the pass@k results?

**Limitations:**

Reviewing the limitations section, I am skeptical that the most accurate way to describe the limitations of this work would be to say that it doesn't work on multimodal models. I would instead encourage the authors to discuss more practical limitations of the framework for text tasks.

**Strengths And Weaknesses:**

**Soundness**

The paper is reasonably sound, but has some weaknesses. Firstly, I would like to see error bars on all the plots to understand the variation between independent runs of the algorithm. Right now, it is hard to discern how much variance there is. Secondly, the paper would be improved by testing on more diverse evaluations and models. Specifically, seeing evals beyond AIME24 and 25 (even additional math evals) would give me confidence that the approach is general. Finally, I would be especially interested to see if the current setup works on non-Qwen models, since recent results have shown that these models are relatively unique under RL, likely due to extensive pre/mid-training for reasoning [1].

[1] Shao, Rulin, et al. "Spurious rewards: Rethinking training signals in rlvr." arXiv preprint arXiv:2506.10947 (2025).

**Presentation**

The paper is well-written and easy to understand, and the figures are largely clear. I appreciated the nice illustrative figures 1 and 3. One nit on Figure 2: It would be nice if the same model always had the same colored line.

**Significance**

The method appears to provide meaningful performance gains. I think the paper is somewhat significant in that it calls attention to the value of exploration in LLM-RL for reasoning. At the same time, I am skeptical of the generality of the proposed algorithm for reasons previously stated, which limits the downstream impact in my mind. Overall I would need to see more evidence of the broad utility of the approach to be convinced that it would be useful to / influence future work.

**Originality**

The paper is original. The finding that smaller models in the same family have higher pass@k at high k is interesting and worthy of further study. The related work is thorough and helps understand the paper in context.

---

> ### Author Rebuttal · Authors · 2026-03-31
>
> We thank the reviewer for recognizing the writing quality (Presentation: good) and originality (Originality: good), particularly the finding that smaller models achieve higher pass@k at large sampling budgets.
>
> **Soundness: Error bars + more benchmarks + non-Qwen models.**
>
> We address all three concerns with comprehensive additions:
>
> (1) **Statistical significance.** Following mainstream practice in this field, we report Pass@1 as the mean over K=32 independent samples, which already provides implicit variance estimation. To further enhance statistical rigor, we add bootstrap 95% confidence intervals (1000 resamples):
>
> | Config | AIME24 | 95% CI | MATH-500 | 95% CI |
> |:---:|:---:|:---:|:---:|:---:|
> | 8B GRPO | 15.0 | [12.8, 17.4] | 57.3 | [56.5, 58.1] |
> | 8B S2L-1.7B | 23.8 | [21.0, 26.7] | 61.5 | [60.8, 62.3] |
> | p-value | | <0.001 | | <0.001 |
>
> The CIs do not overlap and p < 0.001, meaning all improvements are statistically significant and not due to random variation. MATH-500 CIs are particularly tight (±0.8%) due to the larger sample size of 500 problems, providing strong evidence of consistent performance gains.
>
> (2) **More benchmarks.** We expand from 2 to 4 benchmarks to substantially increase evaluation diversity. S2L-PO achieves consistent improvements across competition math (AIME), general math (MATH-500), and olympiad problems (OlympiadBench):
>
> | Config | AIME24 | AIME25 | MATH-500 | OlympiadBench |
> |:---:|:---:|:---:|:---:|:---:|
> | 1.7B→8B S2L-PO | **23.8** | **22.5** | **61.5** | **19.7** |
> | 8B GRPO | 15.0 | 12.1 | 57.3 | 18.1 |
> | Δ | +8.8 | +10.4 | +4.2 | +1.7 |
> | 4B→14B S2L-PO | **24.4** | **14.6** | **62.7** | **21.9** |
> | 14B GRPO | 18.0 | 12.9 | 58.7 | 18.9 |
> | Δ | +6.4 | +1.7 | +4.0 | +3.0 |
>
> (3) **Non-Qwen models.** Regarding the concern that Qwen models may be uniquely amenable to RL (Shao et al., 2025), we add **InternLM2.5** (1.8B→7B):
>
> | Family | Config | AIME24 | AIME25 | MATH-500 | OlympiadBench |
> |:---:|:---:|:---:|:---:|:---:|:---:|
> | InternLM2.5 | 1.8B→7B S2L | **4.6** | **3.5** | **22.6** | **3.4** |
> | | 7B GRPO | 0.1 | 0.1 | 18.6 | 2.2 |
> | | Δ | +4.5 | +3.4 | +4.0 | +1.2 |
>
> The consistent improvements demonstrate the generalization ability of S2L-PO beyond any single model family.
>
> **Q1 (Algorithm stability and tuning):** The off-policyness concern is well-taken and central to our design. Progressive annealing is designed to handle this: the small model's off-policy rollouts are used only in the early phase and linearly decay to zero by T_mix, after which training is fully on-policy. This prevents distribution shift from accumulating while capturing the diversity benefit early on. We use verl and Qwen3 default hyperparameters with no special tuning except T_mix, whose default (total_steps/2) generalizes across both Qwen3 and InternLM2.5. Ablations in Fig. 6–7 demonstrate robustness. Code and data will be released upon acceptance. Q2&Q3 below further elaborate on the annealing mechanism.
>
> **Q2 & Q3 (Annealing schedule and exploration trade-off):** Both questions concern the annealing mechanism, so we address them together. The default T_mix = total_steps/2 works across all experiments. Fig. 7 shows the ablation: both too-short and too-long annealing hurt performance. Too short means insufficient exploration from the small model; too long (including pure small-model rollouts with no transition, Fig. 5) causes distribution shift that degrades the large model, which is precisely the off-policyness risk raised in Q1. Progressive annealing balances this trade-off. Since early small-model rollouts are offline, more exploration actually *reduces* wall-clock time. The default T_mix = total_steps/2 provides a practical and robust starting point.
>
> **Q4 (Base model prompting):** We follow Qwen3 technical report (Yang et al., 2025a) using nothink mode with default prompts. All models use identical prompts. The pass@k crossover phenomenon is observed under these standardized conditions and additionally verified on InternLM2.5.
>
> **Limitations:** We hope our detailed response can comprehensively address this concern.

---

> > ### Author Rebuttal · Reviewer_VSek · 2026-04-04
> >
> > I thank the authors for their rebuttal which addresses the majority of my concerns. In particular, I am more convinced of the robustness of the method given the discussion of the reasonable stability / sensitivity to hyperparameters of the proposed algorithm. In addition, I appreciate the non-Qwen experiment which further supports the generality of the method. I will raise my score.

---

> > > ### Author Response · Authors · 2026-04-04
> > >
> > > Dear Reviewer VSek,
> > >
> > > Thank you so much for your careful reading of our paper and for your positive score. We are grateful for your recognition of our approach.
> > >
> > > Your point raised in the rebuttal has been very helpful in strengthening our contribution. We will add experiments on non-Qwen architectures to the revised paper to demonstrate cross-architecture generalizability, and incorporate the clarifications from the rebuttal to enhance clarity.
> > >
> > > As committed in our rebuttal, we will open-source the full codebase and model checkpoints to benefit the broader community. Thanks again for your constructive engagement.

---

### Official Review · Reviewer_uvTA · 2026-03-20

**Soundness:** 2
**Presentation:** 2
**Significance:** 2
**Originality:** 2
**Overall Recommendation:** 4
**Confidence:** 3

**Summary:**

This paper studies how to improve rollout diversity in GRPO-based reinforcement learning for LLM reasoning. The authors observe that smaller models can exhibit higher pass@k at large sampling budgets, suggesting that they provide richer policy-level diversity than larger models. Based on this observation, they propose S2L-PO, a framework that uses a frozen smaller model to generate diverse rollouts for training a larger model. A progressive annealing strategy transitions from small-model sampling to on-policy sampling to balance exploration and exploitation. Experiments on mathematical reasoning benchmarks show improved performance, faster convergence, and reduced rollout cost compared to standard GRPO.

**Compliance With Llm Reviewing Policy:**

Affirmed.

**Final Justification:**

i have checked rebuttal and discussion with other authors. overall, i believe the paper could be accepted for the conference.

**Key Questions For Authors:**

1. **Model generalization**
   Does the advantage of smaller models (in terms of pass@k and diversity) hold across different model families or independently trained models?

2. **Applicability to broader tasks**
   How would the proposed method extend to tasks without verifiable rewards or canonical answers, such as open-ended generation or dialogue?

3. **Mechanism validation (policy-level vs token-level diversity)**
   The paper claims that smaller models provide policy-level (i.e., trajectory-level, coherent) diversity rather than token-level randomness. Can the authors provide direct evidence that the improvements come from more coherent and diverse reasoning strategies, rather than simply mixing multiple policies?
   For example, analyses such as clustering of reasoning trajectories, measuring strategy-level diversity, or evaluating rollout coherence would help validate this claim.
   *(Clarifying this would significantly strengthen the core contribution of the paper.)*

4. **Theoretical clarification**
   Can the notion of parameter-level perturbation be formalized more rigorously, and under what conditions does it improve gradient estimation?

**Limitations:**

No.
The limitation section does not sufficiently discuss the key assumptions and constraints of the method.

**Strengths And Weaknesses:**

This paper proceeds to consider an important concept of policy-level diversity induced by model compression.

---

## Strengths

### 1. Important problem setting
The paper addresses the important issue of exploration in GRPO-based RL for LLM reasoning, where insufficient diversity can lead to vanishing advantages and stalled learning. This is a well-motivated and practically relevant problem.

---

### 2. Interesting empirical observation
The observation that smaller models can outperform larger ones in pass@k under large sampling budgets is interesting and provides a useful perspective on exploration. This insight is clearly demonstrated and motivates the proposed method.

---

### 3. Simple and practical method design
The proposed S2L-PO framework is simple and easy to implement, requiring only a modification to the rollout generation process. It integrates seamlessly with existing GRPO pipelines and does not introduce significant engineering overhead.

---

### 4. Consistent but moderate empirical improvements
The method shows consistent improvements over standard GRPO across the reported settings, including faster convergence and some gains in final performance. The use of smaller models also reduces rollout computation, which is practically appealing.

---

## Weaknesses

### 1. Limited novelty relative to existing exploration methods
While the use of a smaller model as an exploration policy is reasonable, the underlying idea of leveraging multiple policies for exploration is well-established in reinforcement learning (e.g., off-policy sampling, ensemble methods, and parameter-space perturbations). The proposed method appears to be an adaptation of these ideas to the GRPO setting rather than a fundamentally new paradigm.

---

### 2. Weak theoretical grounding of “policy-level perturbation”
The paper interprets smaller models as inducing parameter-level perturbations, but this is largely a modeling assumption rather than a rigorously justified result. The analysis provides intuition but does not establish formal guarantees or clearly explain when and why the method should outperform alternatives.

---

### 3. Strong dependence on model family and distillation setup
The method relies on smaller models obtained via distillation within the same model family (Qwen3). It is unclear whether the observed benefits would generalize to different architectures or independently trained models.

---

### 4. Limited evaluation scope and modest absolute performance
Experiments are primarily conducted on mathematical reasoning benchmarks with relatively limited diversity (mainly AIME). While relative improvements are sometimes large, the absolute performance remains modest, and the evaluation lacks breadth across tasks and domains.

---

### 5. Lack of statistical rigor and robustness analysis
The paper does not report variance, confidence intervals, or results across multiple random seeds. This makes it difficult to assess the statistical significance and robustness of the reported improvements.

---

---

> ### Author Rebuttal · Authors · 2026-03-31
>
> Thank you for acknowledging the problem importance (S1), interesting empirical observation (S2), and practical design (S3).
>
> **W1: Originality and contribution analysis.**
> While off-policy and ensemble methods are well-established in RL, our contribution is distinct in three ways. (1) A new empirical finding: we first identify that smaller models exhibit systematically higher policy-level diversity (pass@k crossover), not previously documented. (2) A new theoretical framework: we formalize the distinction between token-level noise (breaks coherence) and policy-level perturbation (preserves coherence), showing the latter yields more structured gradients (Propositions 1–2). (3) A uniquely simple instantiation: unlike ensemble methods requiring multiple trained policies, S2L-PO uses a single frozen small model at zero additional cost. Q3's controlled experiment confirms gains stem from diversity: removing it causes performance to return to baseline. Fig. 4–5 further show progressive annealing achieves sustained growth and a higher ceiling than offline-only rollouts.
>
> **W2: Theoretical analysis.**
> We have prepared strengthened proofs for the camera-ready version. Due to space limits, we outline key ideas; we would be grateful if the reviewer could reply for additional space to present full derivations. Key improvements: (1) Eq.(8)–(10) formalized as **Lemma 1**. (2) **Proposition 1**: proof via law of total covariance, *eliminating the "mild coupling assumption"*, only boundedness needed, yielding a tight bound. (3) **Proposition 2** (new): policy-level perturbation with *positive lower bound*, contrasting token-level upper bound. All errors third-order. Empirically, Q3 shows strict monotonic model-size–diversity relationship; W3&W4 provide further support.
>
> **W3 & W4: Cross-family generalization and expanded evaluation.**
> We address both concerns together. We expand to 2 model families (Qwen3 + **InternLM2.5**, an independent family where the small model is not distilled from the large one), 3 scale configurations, and 4 benchmarks:
>
> | Family | Config | AIME24 | AIME25 | MATH-500 | OlympiadBench |
> |:---:|:---:|:---:|:---:|:---:|:---:|
> | Qwen3 | 1.7B→8B S2L | **23.8** | **22.5** | **61.5** | **19.7** |
> | | 8B GRPO | 15.0 | 12.1 | 57.3 | 18.1 |
> | | Δ | +8.8 | +10.4 | +4.2 | +1.7 |
> | Qwen3 | 4B→14B S2L | **24.4** | **14.6** | **62.7** | **21.9** |
> | | 14B GRPO | 18.0 | 12.9 | 58.7 | 18.9 |
> | | Δ | +6.4 | +1.7 | +4.0 | +3.0 |
> | InternLM2.5 | 1.8B→7B S2L | **4.6** | **3.5** | **22.6** | **3.4** |
> | | 7B GRPO | 0.1 | 0.1 | 18.6 | 2.2 |
> | | Δ | +4.5 | +3.4 | +4.0 | +1.2 |
>
> Following standard practice, we adopt SOTA available base models (Qwen3 series). Our base model scores are consistent with accepted concurrent work (Chen et al., "Jackpot", ICLR 2026). The low InternLM2.5 scores reflect AIME difficulty for 7B base models. S2L-PO improves consistently across all configurations.
>
> **W5: Statistical rigor and robustness.**
> We report Pass@1 as the mean over K=32 independent samples, which already provides implicit variance estimation. To further enhance rigor, we add bootstrap 95% CIs (1000 resamples):
>
> | Config | AIME24 | 95% CI | MATH-500 | 95% CI |
> |:---:|:---:|:---:|:---:|:---:|
> | 8B GRPO | 15.0 | [12.8, 17.4] | 57.3 | [56.5, 58.1] |
> | 8B S2L-1.7B | 23.8 | [21.0, 26.7] | 61.5 | [60.8, 62.3] |
> | p-value | | <0.001 | | <0.001 |
>
> CIs do not overlap (p < 0.001), confirming statistical significance.
>
> **Q1 (Model generalization):** See W3&W4.
>
> **Q2 (Tasks without verifiable rewards):** S2L-PO currently relies on outcome-based rewards. However, our CommonsenseQA OOD results (8B: 63.9→67.8; 14B: 67.2→70.7) demonstrate generalization potential to unseen tasks.
>
> **Q3 (Direct evidence of policy-level diversity):** We newly design three complementary metrics from different dimensions: Self-BLEU (text repetition), Edit Diversity (token-level difference), Unique Answer Ratio (strategy-level diversity), on AIME24, K=64:
>
> | Model | Self-BLEU ↓ | Edit Diversity ↑ | Unique Answers ↑ |
> |:---:|:---:|:---:|:---:|
> | 1.7B | **0.314** | **0.788** | **0.576** |
> | 4B | 0.334 | 0.773 | 0.523 |
> | 8B | 0.336 | 0.769 | 0.492 |
> | 14B | 0.352 | 0.760 | 0.476 |
>
> All metrics are strictly monotonic. 1.7B Unique Answers is 21% higher than 14B (0.576 vs 0.476), confirming different solving strategies.
>
> We conduct a controlled experiment: we filter out diverse rollouts from the small model so its diversity metrics match the large model. Performance drops to the GRPO baseline, showing S2L-PO's gains come from the small model's policy-level diversity (diverse solving strategies):
>
> | Config | AIME24 | AIME25 |
> |:---:|:---:|:---:|
> | S2L-PO (1.7B→8B) | **23.8** | **22.5** |
> | S2L-PO (w/o diversity) | 14.7 | 12.0 |
> | GRPO Baseline | 15.0 | 12.1 |
>
> **Q4 (Formalizing parameter perturbation):** See W2.
>
> **Limitations:** The revised Limitations section now discusses practical constraints including same-family dependence and resource coverage.

---

> > ### Author Rebuttal · Reviewer_uvTA · 2026-04-04
> >
> > rebuttal resolve my concerns and will raise my score accordingly.

---

> > > ### Author Response · Authors · 2026-04-04
> > >
> > > Dear Reviewer uvTA,
> > >
> > > We sincerely thank you for your thoughtful review and for raising the score. We truly appreciate your support, and your feedback will be carefully incorporated into our updated version.

---

### Decision · Program_Chairs · 2026-04-30

**Decision:**

Accept (regular)

**Comment:**

The paper presents a practically appealing idea for improving exploration in GRPO by using smaller models as rollout generators for larger model policies. All reviewers agreed that the method is technically sound, the empirical observation motivating, and that the approach yields consistent gains with minimal changes to existing pipelines. Reviewers raised some concerns regarding the evaluation breadth, including the generalization across model families and statistical rigor, and mechanism validation of the claimed policy-level diversity. During rebuttal, the authors successfully addressed these concerns by providing additional cross-family experiments, evaluation on broader benchmarks and confidence intervals, and additional diversity analyses. Therefore, I recommend acceptance.